# Mixed Supervised Object Detection by Transferring Mask Prior and Semantic Similarity

**Yan Liu**,* **Zhijie Zhang**,* **Li Niu**,† **Junjie Chen**, **Liqing Zhang**†
MoE Key Lab of Artificial Intelligence
Department of Computer Science and Engineering
Shanghai Jiao Tong University
{loseover, zzj506506, ustcnewly, chen.bys}@sjtu.edu.cn
zhang-lq@cs.sjtu.edu.cn

## Abstract

Object detection has achieved promising success, but requires large-scale fully-annotated data, which is time-consuming and labor-extensive. Therefore, we consider object detection with mixed supervision, which learns novel object categories using weak annotations with the help of full annotations of existing base object categories. Previous works using mixed supervision mainly learn the class-agnostic objectness from fully-annotated categories, which can be transferred to upgrade the weak annotations to pseudo full annotations for novel categories. In this paper, we further transfer mask prior and semantic similarity to bridge the gap between novel categories and base categories. Specifically, the ability of using mask prior to help detect objects is learned from base categories and transferred to novel categories. Moreover, the semantic similarity between objects learned from base categories is transferred to denoise the pseudo full annotations for novel categories. Experimental results on three benchmark datasets demonstrate the effectiveness of our method over existing methods. Codes are available at https://github.com/bcmi/TraMaS-Weak-Shot-Object-Detection.

## 1 Introduction

With the ubiquitous application of convolutional neural networks and the release of large-scale fully-annotated detection benchmark datasets (*e.g.*, MS COCO [27], ILSVRC Detection [32], and PASCAL VOC [10]), object detection has achieved significant advances in recent years [14, 29, 41]. However, Fully Supervised Object Detection (FSOD) requires massive training images with precise bounding boxes and box-level category labels. Acquiring such full annotations is time-consuming and labor-intensive, which limits the application of FSOD in real-world scenarios. In contrast, weak annotations like image-level category labels are much easier to acquire (*e.g.*, crawl from public websites). Thus, plenty of Weakly Supervised Object Detection (WSOD) methods [22, 30, 15] have been proposed to bridge the gap between full annotation and weak annotation. Due to the lack of box-level annotation, WSOD methods rely on unsupervised proposal generation strategy (*e.g.*, Selective Search [43], Edge Boxes [48]) to extract region proposals. Based on the proposals, [3, 38, 39] formulated WSOD task as Multiple Instance Learning (MIL) problem, which treats each image as a bag or multiple bags of proposals. With known image-level labels and unknown box-level labels, we can infer the box-level label of each proposal with an MIL classifier. However, WSOD methods usually only localize the most discriminative region and confuse objects with co-occurring distractors, so the performance gap between FSOD and WSOD is still very large.

---

*Equal contribution
†Corresponding author

It is worth noting that WSOD ignores the existence of fully-annotated object detection datasets. By taking this into consideration, another learning paradigm using mixed supervision, *i.e.*, full annotations for a set of categories and weak annotations for another set of categories, has been explored by [18, 45, 5, 26]. This learning paradigm has inconsistent names like mixed/cross-supervised object detection in previous literature [26, 5]. However, "mixed/cross-supervised" does not emphasize cross-category transfer and could be easily confused with "semi/omni-supervised". Inspired by weak-shot learning [4], which transfers knowledge from fully-annotated base categories to weakly-annotated novel categories, we refer to this learning paradigm as Weak-SHot Object Detection (WSHOD). Assuming the existence of an off-the-shelf dataset with fully-annotated base categories (precise bounding boxes and box-level labels), with the goal to detect the objects of novel categories, we can collect weakly-annotated data (only image-level labels) for novel categories. Fully-annotated base categories and weakly-annotated novel categories have no overlap. We refer to the dataset with base (*resp.*, novel) categories as source (*resp.*, target) dataset, since our goal is utilizing base categories to help improve the detection performance on novel categories.

Under this paradigm, the key problem is what and how to transfer from fully-annotated base categories to weakly-annotated novel categories. Existing works [26, 45] still follow the typical framework of WSOD, that is, feeding proposals into an MIL classifier. Due to the availability of box-level annotations of base categories, they train a CNN backbone with a region proposal network (RPN) to generate proposals. Since the objects of different categories share many common characteristics (*e.g.*, closed contour), objectness is supposed to be transferrable across different categories. Therefore, the trained network could also generate proposals for novel categories, which will be sent to the MIL classifier.

In this paper, beyond class-agnostic objectness, we consider another two targets to be transferred: mask prior and semantic similarity. Since both base categories and novel categories have image-level labels, we can obtain the coarse semantic masks (*e.g.*, CAM [46]) for each image. We conjecture that the coarse semantic mask could provide strong guidance for detecting objects, so we attempt to integrate coarse semantic mask into our CNN backbone. In particular, we employ the image-level labels to train a classifier to derive CAM [46], and append the derived CAM to the feature map in CNN backbone. Because CAM provides the rough probability that each pixel belongs to a certain category, such mask prior information combined with the feature map could greatly enhance the ability to locate and identify candidate bounding boxes. By saying "transfer mask prior", we mean that the ability to detect objects based on mask prior and feature map can be transferred from base categories to novel categories, which will help detect the objects of novel categories.

Besides, following the iterative training strategy in [45], we progressively mine the pseudo bounding boxes of novel categories, during which the noisy bounding boxes are filtered out. The pseudo bounding boxes with correct (*resp.*, incorrect) pseudo labels assigned by the MIL classifier are referred to as inliers (*resp.*, outliers). We argue that semantic similarity, whether two bounding boxes belong to the same category or not, is class-agnostic. Then, we attempt to transfer semantic similarity from base categories to help identify the outliers of novel categories. Specifically, we train a similarity network based on the ground-truth bounding boxes of base categories to verify whether two bounding boxes belong to the same category. We divide the pseudo bounding boxes of novel categories into batches and the pseudo bounding boxes in each batch have the same pseudo label. Within each batch, we apply the trained similarity network to all pairs of pseudo bounding boxes to compute their semantic similarities, and then average the similarities for each one. If one batch is dominated by inliers, an inlier should be close to most other pseudo bounding boxes and its average similarity should be high. Thus, we can use average similarity to distinguish outliers and inliers.

In summary, the transferred mask prior could help obtain better candidate bounding boxes, while the transferred semantic similarity could help discard the noisy pseudo bounding boxes. We conduct extensive experiments on three datasets. The results demonstrate that our proposed approach with transferred mask prior and semantic similarity can significantly boost the performance of weak-shot object detection. Considering that the key of our method is *Tra*nsferring *Ma*sk prior and *S*imilarity, we dub our method as *TraMaS*. Our main contributions are as follows:

- We propose a novel approach named TraMaS for weak-shot object detection (WSHOD). Besides class-agnostic objectness, we also propose to transfer mask prior and semantic similarity.

- We integrate CAM derivation into our network, which can help generate better candidate bounding boxes with mask prior.

- We employ a similarity network to learn semantic similarity, which can be used to filter out noisy pseudo bounding boxes.

- Our TraMaS method outperforms all state-of-the-art WSOD and WSHOD methods on three benchmark datasets.

## 2 Related Work

### 2.1 Weakly Supervised Object Detection

Weakly supervised object detection (WSOD) proposes to learn a detector with only image-level category labels. Owing to the scarcity of bounding boxes, WSOD tends to generate candidate proposals using unsupervised proposal generation strategies such as Edge Boxes [48] and Selective Search [43]. The final object results are screened from these candidate proposals. Essentially, WSOD can be cast as an image classification problem with multi-instance learning (MIL) classifier. [3] proposed an end-to-end weakly supervised deep detection network which employs two data streams to select proposals and perform region-level classification simultaneously. Following [3], previous WSOD methods can be categorized into two groups according to whether using iterative refinement.

On the one hand, [8] introduced weak cascaded convolutional networks by utilizing Class Activation Map (CAM) and weakly supervised segmentation to improve the quality of generated proposals. [2] minimized the difference between an annotation-aware conditional distribution and an annotation-agnostic prediction distribution, with a dissimilarity coefficient based WSOD framework. [30] proposed a unified instance-aware and context-focused weakly supervised learning architecture to tackle the challenges of WSOD. On the other hand, some methods adopted iterative refinement. [38] unified MIL and multi-stage instance classifiers into a single deep network with online instance classifier refinement. The subsequent work [39] introduced proposal clusters to [38]. [21] explored attention regularization for WSOD training and proposes comprehensive attention self-distillation. Despite the fact that these methods have achieved promising results, because of the scarcity of ground-truth bounding boxes, the localization of objects is still far from satisfactory. In this paper, we follow the second group of methods and adopt an iterative training strategy.

### 2.2 Weak-shot Object Detection

Due to the lack of box-level annotations, WSOD heavily relies on the quality of candidate proposals, and the learned detector can not refine the proposals. Thanks to the off-the-shelf fully-annotated object detection datasets, weak-shot object detection (WSHOD) proposes to leverage the fully-annotated dataset as source dataset to improve the performance of WSOD on weakly-annotated data. Existing WSHOD methods [42, 16, 36] mainly focus on mining various types of knowledge in the base categories and transferring them to novel categories. Many methods [31, 35, 9] learn an appearance model from base categories and transfer it to novel categories. Previous results have proved that class-invariant knowledge can effectively alleviate the problem that WSOD cannot accurately locate objects. For example, [18], [40] and [24] transferred the difference between classifier and detector from base category to novel category. [26] learned class-invariant objectness from base categories with an adversarial domain classifier. With the help of learned objectness knowledge, this method can distinguish objects and distractors, and thus improves the ability to reject distractors in novel categories. In [45], Zhong et al. employed one-class universal detector learned from base category to provide proposals for the MIL classifier, which are both used to mine pseudo ground-truth for iterative refinement. The method in [5] exploited the spatial correlation between high-confidence bounding boxes output by the MIL classifier. In this paper, besides objectness and regression ability, we additionally explore two transfer targets: mask prior and semantic similarity.

Besides weak-shot object detection, weak-shot learning paradigm (transferring knowledge from fully-annotated base categories to weakly annotated novel categories) has also been studied in other computer vision tasks like classification [4], semantic segmentation [47], and instance segmentation [19, 25]. Although transferring similarity has been investigated in [4, 47], transferring mask prior remains unexplored due to the uniqueness of object detection task.

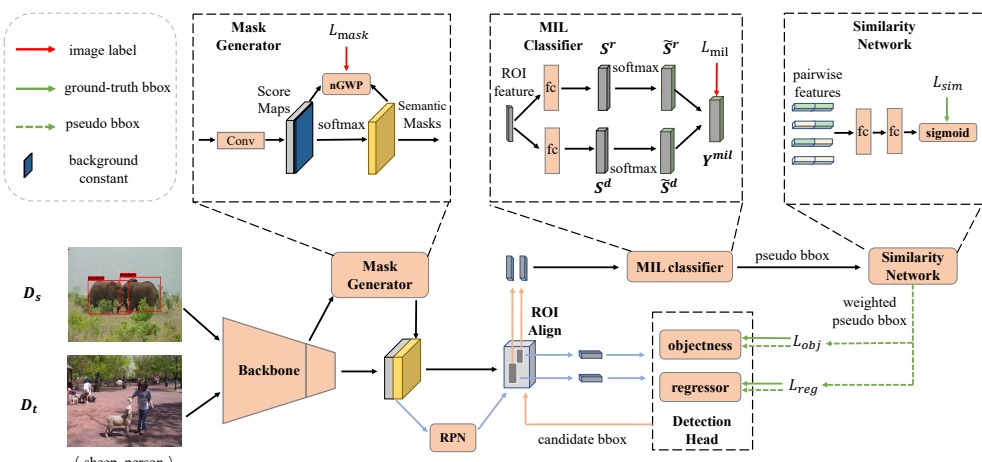

Figure 1: Our network is built upon an object detection network (backbone, RPN, and detection head) and an MIL classifier. We unify a mask generator with the object detection network to provide mask prior information. The similarity network is trained to predict semantic similarity to denoise pseudo bounding boxes. Both the source dataset $D_s$ and target dataset $D_t$ are involved in the whole network training. During testing, we use the object detection network, mask generator, and MIL classifier. Note that the blue (*resp.*, orange) arrow line indicates the data stream of proposals (*resp.*, candidate bounding boxes).

## 3 Our Method

In this paper, our target is to improve the performance of object detection on weakly-annotated novel categories with the help of fully-annotated base categories. Under this paradigm, we have access to source dataset $D_s$ and target dataset $D_t$. $D_s$ has full annotations for base categories (*i.e.*, precise bounding boxes and box-level category labels) while $D_t$ only has weak annotations (image-level category labels) for novel categories. Base categories and novel categories have no overlap. Our method is trained with a mixture of $D_s$ and $D_t$, and evaluated on the test set of novel categories.

### 3.1 Object Detection Network and MIL Classifier

Our basic framework is illustrated in Figure 1. The architecture is composed of an object detection network and an MIL classifier. The object detection network is built upon Faster-RCNN [29] and comprised of a backbone, a Region Proposal Network (RPN), and a detection head. Specifically, we employ ResNet50 [17] as backbone and RPN to generate proposals. The region features of generated proposals are sent to the detection head, which consists of an objectness predictor and a box regressor. Both objectness predictor and box regressor are a fully connected layer. The objectness predictor is used to predict the objectness of a proposal. The objectness label of a proposal is 1 if its IoU with a ground-truth bounding box is above the preset threshold and 0 otherwise. The box regressor is used to regress the proposal to a candidate bounding box. The loss functions of objectness predictor and box regressor are formulated as follow,

$$L_{obj} = -\sum_i \hat{o}_i log o_i + (1 - \hat{o}_i) log(1 - o_i), \quad L_{reg} = \sum_i L_1(b_i, \hat{b}_i), \quad (1)$$

where $L_1$ indicates smooth $L_1$ loss [13]. $o_i$ (*resp.*, $\hat{o}_i$) and $b_i$ (*resp.*, $\hat{b}_i$) mean the predicted (*resp.*, ground-truth) objectness label and regression value of the $i$-th proposal. Following [45], we filter out the proposals with low objectness using the threshold $0.05$ and refer to the remaining proposals as candidate bounding boxes. At the beginning of training process, we supervise the object detection network with the ground-truth bounding boxes in the source dataset. Since objectness has proved to be class-agnostic [26, 45], the object detection network can also generate reasonable candidate bounding boxes for novel categories.

The region features of candidate bounding boxes are fed into an MIL classifier from WSDDN [3]. Note that the MIL classifier only utilizes the target dataset and focuses on novel categories. As shown in Figure 1, the MIL classifier has two branches (*i.e.*, classification branch and detection branch), which are almost the same except the regularization operation. Each branch has two fully connected layers and one activation layer. Given the region features of $N$ candidate bounding boxes, the classification (*resp.*, detection) branch outputs the score matrix $\mathbf{S}^r \in \mathbb{R}^{C_n \times N}$ (*resp.*, $\mathbf{S}^d \in \mathbb{R}^{C_n \times N}$, where $C_n$ is the number of novel categories). Softmax is applied along different dimensions of $\mathbf{S}^r$ and $\mathbf{S}^d$ to obtain category-aware score matrix $\tilde{\mathbf{S}}^r$ and proposal-aware score matrix $\tilde{\mathbf{S}}^d$. Intuitively, $\tilde{\mathbf{S}}^r$ is class score matrix for each proposal while $\tilde{\mathbf{S}}^d$ indicates the contribution of each proposal to classes. The final score matrix $\tilde{\mathbf{S}}$ is the multiplication of $\tilde{\mathbf{S}}^r$ and $\tilde{\mathbf{S}}^d$. Then, $\tilde{\mathbf{S}}$ are summed along the proposal dimension to obtain image-level classification score. By denoting each entry in $\mathbf{S}$ (*resp.*, $\tilde{\mathbf{S}}$) as $s_{c,i}$ (*resp.*, $\tilde{s}_{c,i}$), the above procedure can be formulated as

$$\tilde{s}^r_{c,i} = \frac{e^{s^r_{c,i}}}{\sum_{k=1}^{C_n} e^{s^r_{k,i}}}, \quad \tilde{s}^d_{c,i} = \frac{e^{s^d_{c,i}}}{\sum_{k=1}^{N} e^{s^d_{c,k}}}, \quad \tilde{s}_{c,i} = \tilde{s}^r_{c,i} \tilde{s}^d_{c,i}, \quad y^{mil}_c = \sum_{i=1}^{N} \tilde{s}_{c,i}, \quad L_{mil} = L_{BCE}(y^{mil}_c, \hat{y}_c),$$
(2)

where $L_{mil}$ is the binary cross-entropy loss, which uses image-level category labels $\hat{y}_c$ of novel categories to supervise the final classification scores. Based on the final prediction scores $\tilde{\mathbf{S}}$, we filter out the candidate bounding boxes with the threshold $0.8$. Then, we assign pseudo category labels to the remaining pseudo bounding boxes based on category-aware prediction scores $\tilde{\mathbf{S}}^r$.

After training the MIL classifier, we can mine the pseudo bounding boxes of novel categories from both source and target datasets. These pseudo bounding boxes can be added to supervise the object detection network to detect the objects of novel categories better. Next, we explore two targets to be transferred from base categories to novel categories: mask prior in Section 3.2 and semantic similarity in Section 3.3.

## 3.2 Incorporating Mask Prior into Object Detection Network

Provided with image-level labels, we can obtain the coarse semantic masks (*e.g.*, CAM) of images. These coarse masks provide the rough probability that each pixel belongs to a certain category, which may enhance the ability of the object detection network to locate and identify objects. Previously, some WSOD methods [6, 37, 34, 11, 12] utilize coarse masks (*e.g.*, CAM) to assist in detecting objects. Besides, many weakly supervised semantic segmentation methods [20, 23] take the coarse mask as a baseline and propose different refinement strategies to obtain more accurate semantic masks. In this paper, we unify semantic mask generation and object detection into one network.

Based on the object detection network, we introduce an additional mask generator which is fed with the output feature from the penultimate layer of backbone, as shown in Figure 1. Inspired by [1], the mask generator is composed of three convolutional layers and a normalised Global Weighted Pooling (nGWP). We assume the pixel-wise output of convolutional layers in channel $c$ (channel $c$ corresponds to category $c$) is $p_{c,i,j}$, and then add a background channel with constant value to $p_{c,i,j}$. Further we compute softmax to obtain the semantic mask $m_{c,i,j}$. The predicted image-level score $y_c$ and the multi-label soft-margin loss $L_{mask}$ [28] of mask generator are defined as follow,

$$y_c = \frac{\sum_{i,j} p_{c,i,j} m_{c,i,j}}{\epsilon + \sum_{i,j} m_{c,i,j}}, \quad L_{mask} = -\frac{1}{C} \sum_{c=1}^{C} \hat{y}_c log(\frac{1}{1+e^{-y_c}}) + (1 - \hat{y}_c) log(\frac{e^{-y_c}}{1+e^{-y_c}}), \quad (3)$$

where $C$ is the total number of categories and $\epsilon$ is set as $1.0$. $\hat{y}_c = 1$ if the image contains the category $c$ and $\hat{y}_c = 0$ otherwise.

In contrast to WSOD methods [8, 44] which directly select positive proposals based on the mask, we append the coarse mask to the feature map in the backbone to acquire a mask-enhanced feature map, which can provide prior information for objectness prediction and box regression. Based on the mask-enhanced feature map, we can train a well-performing detection network with a fully-annotated source dataset to generate faithful proposals for base categories. We conjecture that the ability of using mask prior to facilitate object detection can be transferred from base categories to novel categories. Thus, the enhanced detection network is expected to produce better candidate bounding boxes for both base and novel categories.

### 3.3 Semantic Similarity Transfer

Recall that based on the output scores of the MIL classifier, we can mine pseudo bounding boxes of novel categories. However, the labels of pseudo bounding boxes could still be inaccurate. We refer to the bounding boxes with accurate (*resp.*, inaccurate) labels as inliers (*resp.*, outliers). To mitigate the adverse effect of outliers, we attempt to use semantic similarity to recognize the outliers, in which semantic similarity indicates whether two bounding boxes belong to the same category. Specifically, we train a Similarity Network (SimNet) on base categories to predict the semantic similarity between two bounding boxes. Our SimNet consists of two fully connected layers followed by a sigmoid activation layer. We first construct a batch of the region features $\mathbf{F} \in \mathbb{R}^{B \times d}$ from the ground-truth bounding boxes of base categories, in which $B$ is the batch size and $d$ is the feature dimension. we reorganize $\mathbf{F}$ to generate a batch of pairwise features $\tilde{\mathbf{F}} \in \mathbb{R}^{B^2 \times 2d}$. The pairwise features are fed into SimNet and the predicted similarity scores $\mathbf{A}$ are supervised by the ground-truth similarities $\hat{\mathbf{A}}$. Specifically, if a pair of bounding boxes belong to the same category, the corresponding entry in $\hat{\mathbf{A}}$ is 1. Otherwise, the corresponding entry in $\hat{\mathbf{A}}$ is 0. We adopt binary cross-entropy loss for SimNet:

$$L_{sim} = L_{BCE}(\mathbf{A}, \hat{\mathbf{A}}). \tag{4}$$

Considering unbalanced similar and dissimilar pairs, the standard batch sampling strategy (*i.e.*, sampling from the datasets randomly) will cause severe imbalance issues. To this end, we adopt a sampling strategy adapted to our setting. At first, we select $K$ categories from all categories at random. Then we sample $M$ bounding boxes from each selected category randomly to construct a batch, leading to a relatively balanced batch as the input for SimNet. In our experiments, we set both $K$ and $M$ to 8, so $B = K \times M = 64$.

After training SimNet on the ground-truth bounding boxes of base categories, we apply the trained SimNet to the pseudo bounding boxes of novel categories. At each time, we randomly sample $M$ pseudo bounding boxes from the same novel category to construct a batch. Then, we predict the similarity between pairs of bounding boxes. The similarity score between the $i$-th and $j$-th bounding boxes is $a_{i,j}$. For the $i$-th bounding box, we calculate the average of similarity scores between this bounding box and the other bounding boxes:

$$w_i = \frac{1}{M} \sum_{j=1}^{M} \frac{a_{i,j} + a_{j,i}}{2}. \tag{5}$$

For the bounding boxes with the same pseudo label, we conjecture that inliers should be dominant and thus the average similarities of inliers should be higher than those of outliers. Therefore, we use the average similarity $w_i$ as the assigned weight to each bounding box, leading to the following weighted losses:

$$L_{obj}^{w} = -\sum_{i} w_i(\hat{o}_i log(o_i) + (1 - \hat{o}_i)log(1 - o_i)), \quad L_{reg}^{w} = \sum_{i} w_i L_1(b_i, \hat{b}_i), \tag{6}$$

which are basically the same as Eqn. (1) except the weights $w_i$. By assigning higher weights to inliers, the inliers would contribute more to learning a robust classifier.

### 3.4 Iterative Training Strategy

Similar to [45], we train the whole framework iteratively. In each iteration, we execute the following three steps. The whole algorithm is summarized in Algorithm 1.

In the first step, we train the object detection network with mask generator based on the ground-truth bounding boxes of base categories in the source dataset. The loss function of the first step is as follows,

$$L_{step1} = L_{obj}^{w} + \gamma L_{reg}^{w} + \alpha L_{mask}, \tag{7}$$

where $\alpha$ and $\gamma$ are hyper-parameters set as 0.1 and 1.0 via cross-validation, respectively. In the first iteration, weights $w_i$ in Eqn. (7) are all set as 1.

In the second step, we use the object detection network trained in the first step to generate candidate bounding boxes for the target dataset. We train the MIL classifier based on the candidate bounding

---
**Algorithm 1** Iterative Training Strategy
---
**Input:** source dataset $D_s$, target dataset $D_t$, number of refinement iterations $T$
**Output:** Object Detection Network (ODN), MIL classifier, mask generator
  1: Train ODN with mask generator based on $D_s$ ;
  2: Utilize ODN to generate candidate bboxes for $D_t$. Train MIL classifier together with backbone and mask generator based on $D_t$;
  3: **for** i=1, 2, $\cdots$ ,T **do**
  4:    Mine pseudo bboxes of novel categories on $D_s$ and $D_t$ with MIL classifier. Freeze backbone and mask generator, train SimNet based on $D_s$ and utilize SimNet to assign weights to pseudo bboxes;
  5:    Refine ODN with mask generator based on $D_s$ and $D_t$;
  6:    Utilize ODN to generate candidate bboxes for $D_t$. Refine MIL classifier together with backbone and mask generator based on $D_t$.
  7: **end for**
---

boxes in the target dataset. Note that when training the MIL classifier, the backbone and mask generator are also updated to learn better region features of candidate bounding boxes. The loss function of the second step is as follows,

$$L_{step2} = L_{mil} + \beta L_{mask}, \tag{8}$$

where $\beta$ is a hyper-parameter set as $0.1$ via cross-validation.

In the third step, we use the MIL classifier trained in the second step to mine the pseudo bounding boxes of novel categories on both source and target dataset. Note that for the source dataset, we remove the pseudo bounding boxes whose IoUs with ground-truth bounding boxes of base categories are larger than $0.1$. We train the SimNet using the region features of ground-truth bounding boxes of base categories, with the following loss:

$$L_{step3} = L_{sim}. \tag{9}$$

Then, we apply the trained SimNet to the pseudo bounding boxes of novel categories to obtain their weights as described in Section 3.3. Finally, we merge the weighted pseudo bounding boxes of novel categories with the ground-truth bounding boxes of base categories as the supervision of the first step for the next iteration.

## 4  Experiments

### 4.1  Datasets

In our experiments, we introduce our used source dataset and target dataset separately. The source (*resp.*, target) dataset is fully-annotated (*resp.*, weakly-annotated).

**Source Dataset:** Following [45], we investigate COCO 2017 detection dataset [27] as our source dataset. COCO 2017 is composed of an official train/validation split with 118287 and 5000 images with 80 categories covering the 20 categories in VOC. To reduce the impact of novel categories in the source dataset, we follow [45] and remove all the images that have novel categories, *i.e.*, 20 categories in VOC. After that, we denote the resulting COCO 2017 as COCO-60 which consists of 21987 training images and 921 validation images. Both the training set and validation set are merged together as the source dataset.

Besides, we choose ILSVRC 2013 detection dataset [32] as another source dataset. The 200-category dataset, including the 20 categories in VOC, is split into training (395909 images) and validation (20121 images) sets. Similar to COCO-60, we remove the images which contain novel categories and obtain 143905/6229 train/val images. We denote the total 150134 images as ILSVRC-179 (*water bottle* and *wine bottle* in ILSVRC while *bottle* in VOC). COCO-60 and ILSVRC-179 are used as the source dataset separately in different experiments.

**Target Dataset:** We take Pascal VOC 2007 [10] as the target dataset, which is split into trainval (5011 images) and test (4952 images) sets. Following [45], we leverage the trainval set as our training set and evaluate our method on the test set. All 20 categories in VOC are treated as novel categories.

Table 1: mAP comparison with state-of-the-arts on VOC-20 test set. We re-train a Faster RCNN with mined boxes and denote it as "+distill". Unless marked, the backbones of these methods are VGG16. * indicates that the backbone is ResNet50 instead. '(single-scale)' means single-scale testing.

| Method | aero | bike | bird | boat | bootle | bus | car | cat | chair | cow | table | dog | horse | mbike | pers. | plant | sheep | sofa | train | tv | mAP |
|---|---|---|---|---|---|---|---|---|---|---|---|---|---|---|---|---|---|---|---|---|---|
| **WSOD:** | | | | | | | | | | | | | | | | | | | | | |
| WSDDN [3] | 46.4 | 58.3 | 35.5 | 25.9 | 14.0 | 66.7 | 53.0 | 39.2 | 8.9 | 41.8 | 26.6 | 38.6 | 44.7 | 59.0 | 10.8 | 17.3 | 40.7 | 49.6 | 56.9 | 50.8 | 39.3 |
| WCCN [8] | 49.5 | 60.6 | 38.6 | 29.2 | 16.2 | 70.8 | 56.9 | 42.5 | 10.9 | 44.1 | 29.9 | 42.2 | 47.9 | 64.1 | 13.8 | 23.5 | 45.9 | 54.1 | 60.8 | 54.5 | 42.8 |
| OICR [38] | 58.0 | 62.4 | 31.1 | 19.4 | 13.0 | 65.1 | 62.2 | 28.4 | 24.8 | 44.7 | 30.6 | 25.3 | 37.8 | 65.5 | 15.7 | 24.1 | 41.7 | 46.9 | 64.3 | 62.6 | 41.2 |
| PCL [39] | 54.4 | 69.0 | 39.3 | 19.2 | 15.7 | 62.9 | 64.4 | 30.0 | 25.1 | 52.5 | 44.4 | 19.6 | 39.3 | 67.7 | 17.8 | 22.9 | 46.6 | 57.5 | 58.6 | 63.0 | 43.5 |
| Ren et al. [30] | 68.4 | 77.7 | 57.6 | 27.8 | 29.5 | 68.4 | 74.6 | 66.9 | 32.2 | 73.2 | 48.0 | 45.2 | 54.4 | 73.7 | 24.9 | 29.5 | 64.1 | 53.9 | 65.5 | 65.2 | 55.0 |
| Ren et al. [30]+distill | 66.4 | 69.1 | 58.9 | 32.5 | 27.6 | 71.5 | 73.1 | 66.2 | 32.8 | 75.4 | 47.4 | 53.7 | 63.3 | 71.7 | 34.8 | 28.5 | 57.4 | 54.7 | 62.5 | 67.1 | 55.7 |
| CASD [21] | 66.4 | 82.2 | 56.6 | 33.1 | 27.2 | 76.0 | 54.3 | 69.4 | 33.5 | 59.9 | 57.4 | 52.3 | 68.4 | 73.1 | 26.7 | 28.1 | 68.3 | 55.2 | 73.4 | 71.2 | 56.6 |
| CASD [21]+distill | 66.6 | 81.3 | 58.4 | 33.5 | 31.6 | 75.7 | 55.2 | 68.3 | 36.8 | 59.5 | 61.0 | 52.9 | 65.4 | 72.0 | 29.1 | 29.4 | 65.7 | 54.2 | 74.5 | 70.7 | 57.1 |
| **WSHOD:** | | | | | | | | | | | | | | | | | | | | | |
| MSD [26] | 44.6 | 29.1 | 59.4 | 32.8 | 34.0 | 72.1 | 68.5 | 75.8 | 16.4 | 71.2 | 15.6 | 76.6 | 60.7 | 55.7 | 33.8 | 17.2 | 67.2 | 48.6 | 57.5 | 58.6 | 49.8 |
| Chen et al. [5]* | 56.3 | 38.8 | 68.3 | 47.8 | 49.8 | 77.6 | 75.0 | 76.0 | 32.2 | 75.6 | 22.1 | 80.5 | 75.4 | 59.1 | 43.2 | 15.0 | 67.3 | 59.5 | 72.1 | 67.6 | 58.0 |
| Zhong et al. [45]* | 64.8 | 50.7 | 65.5 | 45.3 | 46.4 | 75.7 | 74.0 | 80.1 | 31.3 | 77.0 | 26.2 | 79.3 | 74.8 | 66.5 | 57.9 | 11.5 | 68.2 | 59.0 | 74.7 | 65.5 | 59.7 |
| Zhong et al. [45]+distill | 62.6 | 56.1 | 64.5 | 40.9 | 44.5 | 74.4 | 76.8 | 80.5 | 30.6 | 75.4 | 25.5 | 80.9 | 73.4 | 71.0 | 59.1 | 16.7 | 64.1 | 59.5 | 72.4 | 68.0 | 59.8 |
| Zhong et al. [45]+distill* | 65.5 | 57.7 | 65.1 | 41.3 | 43.0 | 73.6 | 75.7 | 80.4 | 33.4 | 72.2 | 33.8 | 81.3 | 79.6 | 63.0 | 59.4 | 10.9 | 65.1 | 64.2 | 72.7 | 67.2 | 60.2 |
| **Ours:** | | | | | | | | | | | | | | | | | | | | | |
| Ours*(single-scale) | 65.6 | 53.7 | 67.4 | 47.2 | 46.9 | 76.3 | 76.6 | 81.7 | 33.0 | 76.9 | 29.3 | 80.9 | 76.8 | 66.2 | 61.1 | 12.6 | 65.8 | 58.9 | 74.4 | 66.7 | 60.9 |
| Ours | 66.4 | 53.7 | 68.0 | 48.8 | 47.8 | 76.2 | 78.2 | 81.8 | 34.2 | 77.3 | 29.0 | 81.8 | 78.1 | 65.4 | 64.0 | 14.6 | 66.9 | 60.9 | 75.8 | 68.7 | 61.9 |
| Ours* | 66.5 | 58.7 | 68.3 | 47.7 | 47.0 | 76.3 | 78.0 | 81.1 | 33.9 | 77.8 | 30.9 | 80.1 | 78.0 | 66.2 | 63.0 | 15.1 | 69.2 | 60.2 | 76.1 | 68.1 | 62.1 |
| Ours+distill | 67.8 | 59.9 | 67.9 | 48.9 | 47.5 | 75.4 | 78.2 | 79.3 | 33.1 | 76.4 | 32.1 | 78.8 | 77.4 | 68.3 | 63.1 | 18.4 | 70.0 | 59.9 | 76.2 | 69.3 | 62.4 |
| Ours+distill* | 68.6 | 61.1 | 69.6 | 48.1 | 49.9 | 76.3 | 77.8 | 80.9 | 34.9 | 77.0 | 31.1 | 80.9 | 78.5 | 66.3 | 64.0 | 19.1 | 69.1 | 62.3 | 74.4 | 69.1 | 62.9 |
| **FSOD:** | | | | | | | | | | | | | | | | | | | | | |
| Faster RCNN* | 75.9 | 83.0 | 74.4 | 60.8 | 56.5 | 79.0 | 83.8 | 83.6 | 54.9 | 81.6 | 66.8 | 85.3 | 84.3 | 77.4 | 82.6 | 47.3 | 74.0 | 72.2 | 78.0 | 74.8 | 73.8 |

Table 2: CorLoc comparison with state-of-the-arts on VOC-20 trainval set. We re-train a Faster RCNN with mined boxes and denote it as "+distill". Unless marked, the backbones of these methods are VGG16. * indicates that the backbone is ResNet50 instead. '(single-scale)' means single-scale testing.

| Method | aero | bike | bird | boat | bootle | bus | car | cat | chair | cow | table | dog | horse | mbike | pers. | plant | sheep | sofa | train | tv | Cor. |
|---|---|---|---|---|---|---|---|---|---|---|---|---|---|---|---|---|---|---|---|---|---|
| **WSOD:** | | | | | | | | | | | | | | | | | | | | | |
| WSDDN [3] | 68.9 | 68.7 | 65.2 | 42.5 | 40.6 | 72.6 | 75.2 | 53.7 | 29.7 | 68.1 | 33.5 | 45.6 | 65.9 | 86.1 | 27.5 | 44.9 | 76.0 | 62.4 | 66.3 | 66.8 | 58.0 |
| WCCN [8] | 83.9 | 72.8 | 64.5 | 44.1 | 40.1 | 65.7 | 82.5 | 58.9 | 33.7 | 72.5 | 25.6 | 53.7 | 67.4 | 77.4 | 26.8 | 49.1 | 68.1 | 27.9 | 64.5 | 55.7 | 56.7 |
| OICR [38] | 81.7 | 80.4 | 48.7 | 49.5 | 32.8 | 81.7 | 85.4 | 40.1 | 40.6 | 79.5 | 35.7 | 33.7 | 60.5 | 88.8 | 21.8 | 57.9 | 76.3 | 59.9 | 75.3 | 81.4 | 60.6 |
| PCL [39] | 79.6 | 85.5 | 62.2 | 47.9 | 37.0 | 83.8 | 83.4 | 43.0 | 38.3 | 80.1 | 50.6 | 30.9 | 57.8 | 90.8 | 27.0 | 58.2 | 75.3 | 68.5 | 75.7 | 78.9 | 62.7 |
| Ren et al. [30] | 86.7 | 62.4 | 71.8 | 51.6 | 16.4 | 72.1 | 71.9 | 60.8 | 35.7 | 78.8 | 24.3 | 46.0 | 54.8 | 84.3 | 19.4 | 30.8 | 76.3 | 35.8 | 74.5 | 55.6 | 55.5 |
| Ren et al. [30]+distill | 86.2 | 55.8 | 78.8 | 44.7 | 15.9 | 68.8 | 81.8 | 62.2 | 32.2 | 78.3 | 26.3 | 54.7 | 58.0 | 76.9 | 28.6 | 32.9 | 76.1 | 36.5 | 77.2 | 59.6 | 56.6 |
| CASD [21] | 81.3 | 80.1 | 70.2 | 47.7 | 42.5 | 75.4 | 76.3 | 51.0 | 79.8 | 42.8 | 40.2 | 31.1 | 47.5 | 87.9 | 24.2 | 43.2 | 74.9 | 70.1 | 75.5 | 60.4 | 60.1 |
| CASD [21]+distill | 83.4 | 79.7 | 75.1 | 46.9 | 42.7 | 76.5 | 72.5 | 53.6 | 75.4 | 46.2 | 37.7 | 32.0 | 44.9 | 86.7 | 27.5 | 46.2 | 74.3 | 70.8 | 79.4 | 65.1 | 60.8 |
| **WSHOD:** | | | | | | | | | | | | | | | | | | | | | |
| MSD [26] | 81.5 | 62.1 | 88.2 | 69.9 | 42.1 | 74.4 | 84.1 | 93.9 | 32.8 | 94.1 | 24.9 | 89.2 | 90.5 | 78.3 | 46.8 | 31.4 | 90.3 | 43.1 | 86.9 | 57.8 | 68.1 |
| Chen et al. [5] | 84.5 | 53.7 | 80.7 | 69.5 | 57.0 | 78.5 | 81.8 | 82.2 | 55.0 | 84.9 | 56.4 | 80.6 | 83.6 | 72.2 | 49.2 | 35.3 | 79.6 | 51.9 | 84.0 | 65.9 | 72.4 |
| Zhong et al. [45]* | 87.5 | 64.7 | 87.4 | 69.7 | 67.9 | 86.3 | 88.8 | 88.1 | 44.4 | 93.8 | 31.9 | 89.1 | 92.9 | 86.3 | 71.5 | 22.7 | 94.8 | 56.5 | 88.2 | 76.3 | 74.4 |
| Zhong et al. [45]+distill | 87.9 | 66.7 | 87.7 | 67.6 | 70.2 | 85.8 | 89.9 | 89.2 | 47.9 | 94.5 | 30.8 | 91.6 | 91.8 | 87.6 | 72.2 | 23.8 | 91.8 | 67.2 | 88.6 | 81.7 | 75.7 |
| Zhong et al. [45]+distill* | 85.8 | 67.5 | 87.1 | 68.6 | 68.3 | 85.8 | 90.4 | 88.7 | 43.5 | 95.2 | 31.6 | 90.9 | 94.2 | 88.8 | 72.4 | 23.8 | 98.7 | 66.1 | 89.7 | 76.7 | 75.2 |
| **Ours:** | | | | | | | | | | | | | | | | | | | | | |
| Ours*(single-scale) | 88.9 | 66.5 | 87.3 | 69.2 | 70.6 | 86.2 | 90.3 | 90.6 | 49.5 | 95.5 | 31.6 | 93.7 | 93.5 | 87.4 | 73.6 | 24.9 | 93.5 | 67.3 | 89.6 | 82.7 | 76.6 |
| Ours | 88.4 | 67.4 | 88.0 | 68.3 | 70.3 | 85.7 | 90.0 | 92.4 | 49.9 | 94.7 | 32.0 | 94.1 | 92.6 | 87.5 | 73.2 | 25.3 | 93.8 | 67.7 | 89.7 | 84.2 | 76.8 |
| Ours* | 88.3 | 67.9 | 89.8 | 68.0 | 70.8 | 88.6 | 90.6 | 91.8 | 50.3 | 96.6 | 31.8 | 93.5 | 92.2 | 88.2 | 72.8 | 25.2 | 94.2 | 67.4 | 90.3 | 84.4 | 77.1 |
| Ours+distill | 89.7 | 69.4 | 90.9 | 68.5 | 71.1 | 86.9 | 91.5 | 91.0 | 50.1 | 96.4 | 33.2 | 92.4 | 92.7 | 90.1 | 75.3 | 24.8 | 93.3 | 69.8 | 90.6 | 83.1 | 77.5 |
| Ours+distill* | 90.6 | 67.4 | 89.7 | 70.5 | 72.8 | 86.6 | 91.7 | 89.8 | 51.0 | 96.1 | 34.0 | 93.7 | 94.8 | 90.3 | 73.0 | 26.5 | 95.2 | 68.2 | 89.8 | 83.1 | 77.7 |
| **FSOD:** | | | | | | | | | | | | | | | | | | | | | |
| Faster RCNN* | 99.6 | 96.1 | 99.1 | 95.7 | 91.6 | 94.9 | 94.7 | 98.3 | 78.7 | 98.6 | 85.6 | 98.4 | 98.3 | 98.8 | 96.6 | 90.1 | 99.0 | 80.1 | 99.6 | 93.2 | 94.3 |

Considering that VOC is fully-annotated, we ignore the bounding boxes and only keep the image-level category labels as the supervision for WSHOD and name the processed VOC as VOC-20.

## 4.2 Implementation Details

Our proposed framework is based on Faster RCNN [29] with ResNet-50 [17] pretrained on ImageNet [32] as the backbone. Following [45], we train our model using the SGD optimizer. The learning rate is initialized to $8 \times 10^{-3}$ and reduced to $8 \times 10^{-4}$. The weight decay and momentum are set to $1 \times 10^{-4}$ and 0.9, respectively. The random seed is set to 222. All experiments are conducted on two 24GB TITAN RTX, with a batch size of 16 images. Recall that we train our whole framework in an iterative manner, we conduct four iterations ($T = 4$) following [45]. The threshold used to obtain candidate bounding boxes is 0.05. The threshold used to obtain pseudo bounding boxes of novel categories is 0.8. Following [45], we adopt two evaluation metrics, *i.e.*, mean average precision (mAP) and Correct Localization (CorLoc) [7]. mAP is calculated on VOC-20 test set and CorLoc is calculated on VOC-20 trainval set. Following multi-scale training/testing strategy in previous works [26, 38], we use five image scales $\{480, 576, 688, 864, 1200\}$ for both training and testing unless otherwise specified.

Table 3: Ablation studies on different modules on VOC-20 test set. $B_{-1}$ (*resp.*, $B_{-2}$) indicates that mask is appended to the feature map of the last (*resp.*, penultimate) layer of the backbone. All methods adopt single-scale testing.

| | Mask | | Similarity | | |
|---|---|---|---|---|---|
| | $B_{-1}$ | $B_{-2}$ | Cosine | SimNet | mAP |
| 1 | | | | | 58.2 |
| 2 | ✓ | | | | 59.7 |
| 3 | | ✓ | | | 59.1 |
| 4 | | | ✓ | | 58.8 |
| 5 | | | | ✓ | 59.5 |
| 6 | ✓ | | | ✓ | 60.9 |

Table 4: Result comparison with WSHOD methods when taking ILSVRC-179 as source dataset. Unless marked, the backbones of these methods are VGG16. * indicates that the backbone is ResNet50 instead.

| Methods | mAP | CorLoc |
|---|---|---|
| MSD [26] | 47.5 | 65.3 |
| Chen et al. [5] | 55.1 | 69.3 |
| Zhong et al. [45]* | 56.5 | \ |
| Ours | 57.8 | 74.1 |
| Ours* | 58.3 | 74.8 |

### 4.3 Experiments on COCO-60

In this section, we use COCO-60 as the source dataset and VOC-20 as the target dataset. We compare our TraMaS method with the state-of-the-art approaches on VOC-20, including Weakly-Supervised Object Detection (WSOD) methods WSDDN [3], WCCN [8], OICR [38], PCL [39], Ren et al. [30], and CASD [21], as well as Weak-SHot Object Detection (WSHOD) methods [26, 5, 45]. All baselines take VGG16 as the backbone except [45] which also uses ResNet50. Following [45], we apply distillation strategy (*i.e.*, retrain a Faster RCNN with pseudo labels) to two competitive WSOD baselines (CASD [21] and Ren et al. [30]), the competitive WSHOD baseline [45], and our method. Specifically, distillation means training a full supervised object detector with obtained pseudo bounding boxes of novel categories on COCO-60 and VOC-20. The results of all baselines are copied from [21, 45] except [26, 5] which are reproduced based on our implementation. We use ResNet50 as the backbone by default and also report the results using VGG16. We observe that the results using VGG16 are only slightly worse than those using ResNet50, which is consistent with the observation in [33]. One possible explanation is that the MIL classifier may back-propagate uncertain and inaccurate gradients to backbones while skip connection in ResNet50 can not alleviate this issue. We report mAP and CorLoc in Table 1 and Table 2 respectively. It can be seen that the strongest baseline is [45]. No matter using ResNet50 or VGG16+distill, our TraMaS method significantly outperforms [45].

### 4.4 Experiments on ILSVRC-179

We also conduct experiments by using ILSVRC-179 as the source dataset and VOC-20 as the target dataset. The results are reported in Table 4. Note that since the target dataset is the same, the results of WSOD baselines are the same as in Table 1 and Table 2. The baseline results are copied from [26, 45] except [5] which is reproduced based on our implementation. The results in Table 4 again demonstrate the effectiveness of our method.

### 4.5 Ablation Studies

By taking COCO-60 as the source dataset, we conduct ablation studies to investigate the effectiveness of mask prior and semantic similarity, the results are shown in Table 3. We first evaluate the plain framework without mask prior and similarity weight (row 1). Based on the plain framework in row 1, we append the coarse semantic mask to the last layer in the backbone (row 2). We also try to append the coarse mask to the penultimate layer (row 3). By comparing row 2, 3 and row 1, we can see that incorporating mask prior into the backbone can dramatically increase the performance, *i.e.*, 59.7% and 59.1% *v.s.* 58.2%. The comparison between row 2 and row 3 shows that $B_{-1}$ achieves better performance, so we append the mask prior to the last layer by default.

Based on the plain framework in row 1, we use similarity weights to suppress the outliers (pseudo bounding boxes with inaccurate labels) mined by the MIL Classifier, leading to the results in row 5. By comparing row 1 and row 5, we can see that transferred similarities are effective in identifying the noisy pseudo bounding boxes. Considering that similarity can be calculated in an easier way, we also

remove SimNet and simply calculate cosine similarities based on the region features between two bounding boxes. The results are shown in row 4, which shows that it is useful to train a SimNet to predict similarity.

The last row is our full-fledged method, in which the combination of mask prior and similarity weight achieves further improvement.

### 4.6 Hyper-parameters Analyses

Our method introduces three hyper-parameters, *i.e.*, $\alpha$, $\beta$, and $\gamma$ in Eqn. (7) and (8). Training SimNet also involves two hyper-parameters: $K$ and $M$. The analyses of these hyper-parameters are left to Supplementary due to space limitation. For all the other hyper-parameters, we follow [45] without further tuning.

### 4.7 Size of Source Data

In order to explore the influence of the amount of fully supervised data on weak-shot object detection. We study different scales of datasets as source data and conduct experiments with our method. Specifically, we randomly sample 20% and 50% of COCO-60 as the source dataset and use VOC-20 as target dataset. Based on the setting of 'Ours*(single scale)', we report the results on VOC-20 test set. The final mAP on 20% source data is 59.4% and mAP on 50% source data is 59.8%. It can be seen that as the size of source dataset increases, the performance is improved as well.

### 4.8 Overlap in Base and Novel Categories

In some real-world cases, a few categories may have both fully-annotated data and weakly-annotated data. Therefore, we conduct the experiments with our proposed method under this setting. In our original COCO-60 dataset, we removed all the images which contains the objects of novel categories following [45]. Now among the 20 novel categories, we choose the first 10 novel categories based on the alphabetical order as mixed categories, and treat the remaining 10 categories as novel categories. Then, we only remove the images which contains the objects of 10 novel categories from COCO and name it COCO-70 instead of COCO-60 dataset, so that 10 mixed categories have both fully-annotated data in the source dataset and weakly-annotated data in the target dataset. We conduct experiments with 'Ours+distill*'. The final mAP 69.9% (mixed categories) *v.s.* 62.4% (novel categories) show that the performance on mixed categories are much higher than novel categories thanks to the extra supervision of bounding boxes of mixed categories.

### 4.9 Visualization Results

To better explore the superiority of our method, we visualize some test images, with bounding boxes predicted by different methods on VOC-20. The visualization results can be found in Supplementary. It can be seen that previous WSOD and WSHOD methods (*e.g.*, [45]) tend to focus on the most discriminative regions, while our method is encouraged to detect the whole object and thus locates the objects with higher precision.

## 5 Conclusion

In this work, we have investigated weak-shot object detection by transferring the knowledge learned from the fully-annotated source dataset. Besides the class-invariant objectness explored in previous works, we have also transferred mask prior and semantic similarity. Extensive experiments have proved the superiority of our proposed method.

## Acknowledgements

The work was supported by the National Key R&D Program of China (2018AAA0100704), National Natural Science Foundation of China (Grant No. 61902247), the Shanghai Municipal Science and Technology Major Project (Grant No. 2021SHZDZX0102) and the Shanghai Municipal Science and Technology Key Project (Grant No. 20511100300).

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
