# Supplementary for Mixed Supervised Object Detection by Transferring Mask Prior and Semantic Similarity

**Yan Liu**[*], **Zhijie Zhang**[*], **Li Niu**[†], **Junjie Chen**, **Liqing Zhang**[†]
MoE Key Lab of Artificial Intelligence
Department of Computer Science and Engineering
Shanghai Jiao Tong University
{loseover, zzj506506, ustcnewly, chen.bys}@sjtu.edu.cn
zhang-lq@cs.sjtu.edu.cn

In this supplementary material, we will provide more analyses of mask prior in Section 1 and similarity transfer in Section 2. We will show the visualization results in Section 3 and the performance variance with iteration in Section 4. We will also conduct experiments to mine base categories in the target dataset in Section 5. Besides, the hyper-parameters analyses will be provided in Section 6. Finally, we will discuss the limitations in Section 7.

## 1 Analysis of Mask Prior

As mentioned in Section 3.2 in the main paper, mask prior provides coarse pixel-wise category information to improve the ability of the object detection network to locate and identify objects. Our ablation studies (Table 3 in the main paper) have already proved the advantage of mask prior. To further evaluate the effectiveness of mask prior, we evaluate object detection network with/without mask generator on VOC test set. Considering that the target dataset may contain both base categories and novel categories, in which only novel categories have ground-truth bounding boxes, we evaluate our method on novel categories. Specifically, for both the object detection network with/without mask generator, we utilize regressor to refine proposals from RPN, and simultaneously filter out these proposals with low objectness using threshold $0.05$ to obtain the candidate bounding boxes (as mentioned in Section 3.1 in the main paper). Finally, we calculate recall between candidate bounding boxes and ground-truth bounding boxes of novel categories to access the performance of object detection network. The result with mask generator is $89.6\%$ against $85.3\%$ without mask generator, which demonstrates that mask prior can boost the performance of the object detection network and adapt to novel categories.

Furthermore, we visualize the coarse masks and the corresponding detection results in Section 3 (see Figure 2) to better investigate the effectiveness of mask prior. From Figure 2, we can see that the coarse masks indicate the rough locations of objects which can help the object detection network predict the bounding boxes.

## 2 Analysis of Similarity Transfer

To validate the transferability of our similarity transfer, we evaluate our similarity network trained on COCO-60 trainval set. We treat the similarity prediction task as a binary classification task, in which the binary label 1 (*resp.*, 0) means that two bounding boxes belong to the same category (*resp.*, different categories). We apply a threshold $0.5$ to the predicted similarity score to obtain the binary label. To evaluate the predicted similarities on base categories, we use the removed images

---

[*]Equal contribution
[†]Corresponding author

35th Conference on Neural Information Processing Systems (NeurIPS 2021).

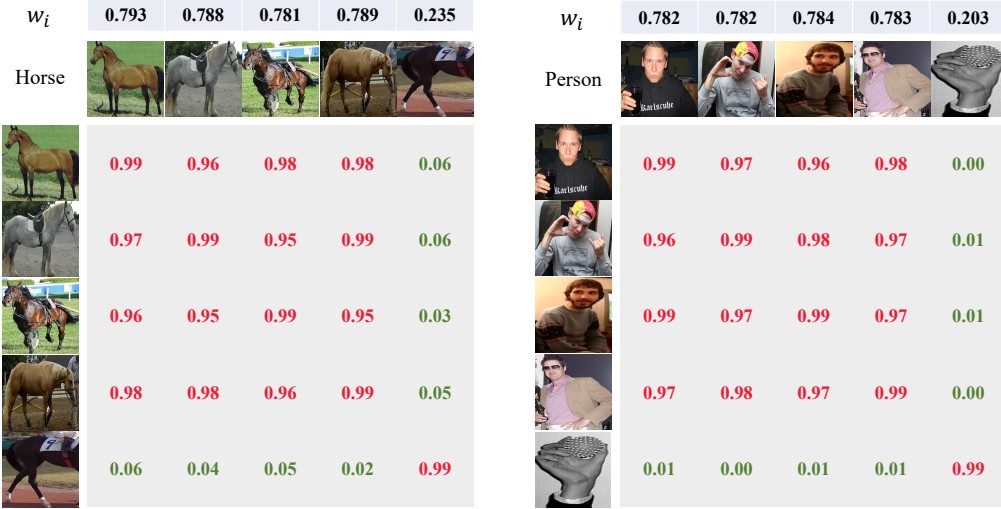

| $w_i$ | 0.793 | 0.788 | 0.781 | 0.789 | 0.235 |
|---|---|---|---|---|---|
| Horse | | | | | |
| | 0.99 | 0.96 | 0.98 | 0.98 | 0.06 |
| | 0.97 | 0.99 | 0.95 | 0.99 | 0.06 |
| | 0.96 | 0.95 | 0.99 | 0.95 | 0.03 |
| | 0.98 | 0.98 | 0.96 | 0.99 | 0.05 |
| | 0.06 | 0.04 | 0.05 | 0.02 | 0.99 |

| $w_i$ | 0.782 | 0.782 | 0.784 | 0.783 | 0.203 |
|---|---|---|---|---|---|
| Person | | | | | |
| | 0.99 | 0.97 | 0.96 | 0.98 | 0.00 |
| | 0.96 | 0.99 | 0.98 | 0.97 | 0.01 |
| | 0.99 | 0.97 | 0.99 | 0.97 | 0.01 |
| | 0.97 | 0.98 | 0.97 | 0.99 | 0.00 |
| | 0.01 | 0.00 | 0.01 | 0.01 | 0.99 |

Figure 1: Visualization of the similarity matrix. The entries in each matrix are the pairwise similarity scores of five randomly selected bounding boxes corresponding to the novel category "*horse*" and "*person*". The upper $w_i$ are the assigned weights, *i.e.*, average similarity scores. The scores below the threshold are in green, whereas the scores above the threshold are in red.

in COCO (mentioned in Section 4.1 in the main paper), which are referred to as COCO/60. We evaluate the performance of the trained similarity network on both base categories of COCO/60 and novel categories of VOC test set. For COCO/60, we calculate the similarities between pairs of region features of ground-truth bounding boxes, in which the region features are extracted from the backbone via ROI align. Then, we gather all region features of each category $c$ and randomly sample the same number of region features from other categories to obtain the stacked features $\mathbf{F} \in \mathbb{R}^{2N_c \times d}$ (where $N_c$ is the number of region features for category $c$ and $d$ is the feature dimension). Next, we reorganize $\mathbf{F}$ to generate pairwise features $\tilde{\mathbf{F}} \in \mathbb{R}^{4N_c^2 \times 2d}$ as mentioned in Section 3.3 in the main paper. Based on the prediction results of pairwise features, we calculate precision, recall, and F1 scores for the binary classification task. For VOC test set, we repeat the above procedure. The precision, recall and F1 scores are summarized in Table 1. We observe that the gap between the performance of similarity network on base categories and novel categories is negligible (*e.g.*, F1 Scores $84.9\%$ *v.s.* $84.1\%$), which indicates that our similarity network generalizes well to novel categories.

Besides, we visualize the similarity score matrix which should accord with our conjecture that outliers will be assigned low scores as claimed in Section 3.3 in the main paper. Specifically, we randomly select five pseudo bounding boxes from two randomly selected novel category "*horse*" and "*person*" and calculate their pairwise semantic similarity scores by applying the trained similarity network. Then, we calculate the average similarity score for each pseudo bounding box as described in Section 3.3 in the main paper. It is well worth noting that the average similarity score will be affected slightly by the number of outliers if batch size increases to a large scale (*e.g.*, 64). Here, we only display semantic similarity among five sampled instances for convenience. As shown in Figure 1, the average similarity scores of inliers (*resp.*, outliers) are relatively higher (*resp.*, lower). These results demonstrate that our similarity network can distinguish outliers and suppress their contribution to network training by assigning lower weights.

## 3 Visualization Results

To qualitatively demonstrate the superiority of our method, we visualize some test images, with bounding boxes predicted by different methods on VOC-20. We qualitatively compare our method

Table 1: Performance test on both COCO/60 and VOC test. "Random" indicates random guess on 20 novel categories.

| Test Dataset | Precision(%) | Recall(%) | F1 Score(%) |
|---|---|---|---|
| Base Categories on COCO/60 | 85.6 | 84.3 | 84.9 |
| Novel Categories on VOC test | 84.8 | 83.5 | 84.1 |
| Random | 26.3 | 50.0 | 34.5 |

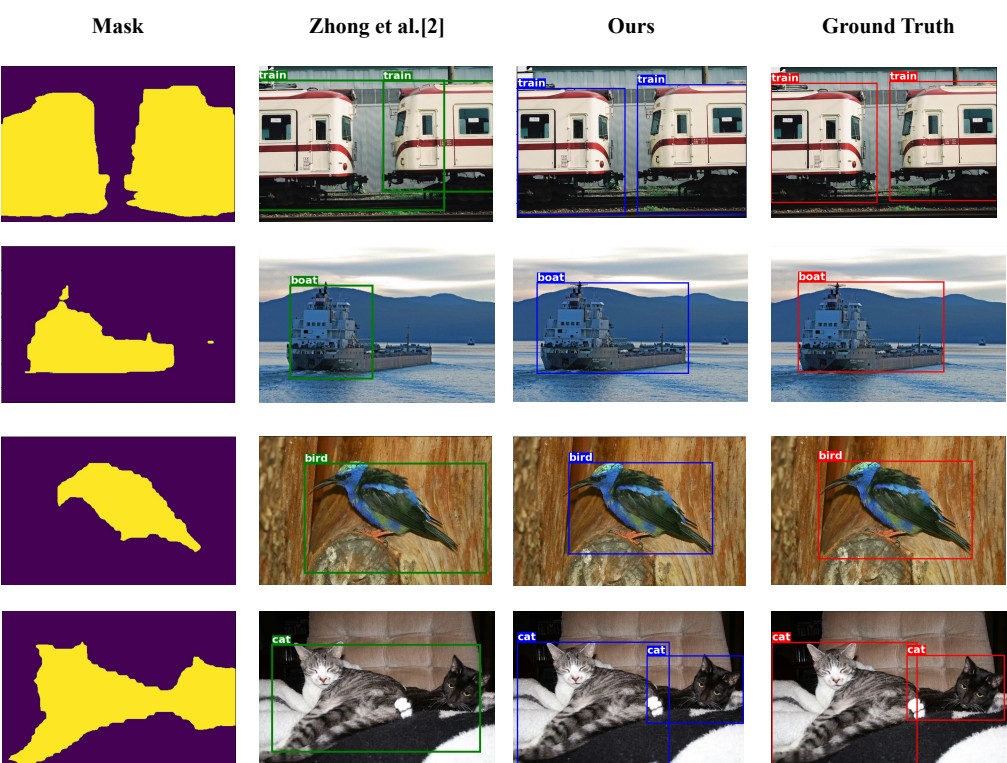

Figure 2: The comparison between the bounding boxes predicted by baseline Zhong et al. [2] and our method. The first column shows the generated mask prior, whereas the second and third columns show the predicted bounding boxes of [2] and our method, respectively. The last column shows the corresponding ground-truth bounding boxes.

with the most competitive baseline Zhong et al. [2]. We also show the coarse masks generated from mask generator.

As shown in Figure 2, the coarse masks can roughly indicate the locations and scales of different objects. Although the mask boundaries are not very accurate, the coarse masks can still help the object detection network locate and identify objects, as discussed in Section 1. From the detection results, we can see that Zhong et al. [2] tends to focus on the most discriminative regions (*e.g.*, the second column in row 2 in Figure 2) or be confused by co-occurring objects (*e.g.*, the second column in row 1 and 4 in Figure 2), while our method can detect the whole objects with the assistance of mask prior and thus locate the objects more accurately.

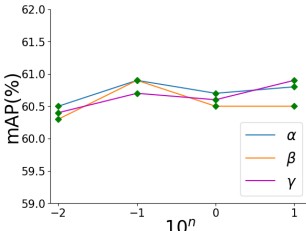 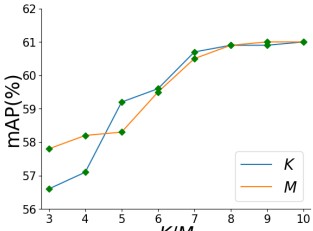 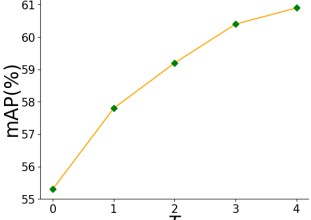

Figure 3: The performance variance of our full-fledged method with various $\alpha$, $\beta$, and $\gamma$ on VOC-20.

Figure 4: The performance variance of our full-fledged method with various $K$ and $M$ on VOC-20.

Figure 5: The performance variance of our full-fledged method after each iteration of refinement.

## 4 Performance of Each Iteration

Following [2], we adopt an iterative training strategy, as described in Section 3.4 in the main paper. Specifically, we gradually mine pseudo bounding boxes in the target dataset using the latest MIL classifier to refine object detection network. To investigate the effectiveness of the iterative training strategy, we report the results of each iteration in Figure 5. It is noticeable that the performance increases stably until iteration 4, which confirms the effectiveness of iterative training.

## 5 Mining Base Categories in Target Dataset

Although we aim to detect objects of novel categories in the target dataset, there also exist objects of base categories in the target dataset, which may affect the training process. To investigate the impact of base categories in the target dataset, we attempt to detect base categories in the target dataset and also use them in the training process. Specifically, we train a fully supervised Faster RCNN with COCO-60 on base categories and use the trained detector to mine pseudo bounding boxes of base categories in the target dataset. Recall that we have three steps in each iteration during our iterative training process as described in Section 3.4 in the main paper. We only need to modify the first step by utilizing the mined pseudo bounding boxes because the bounding boxes of base categories are only required in the first step. We train and refine the object detection network using pseudo bounding boxes of base/novel categories in the target dataset and pseudo (*resp.*, ground-truth) bounding boxes of novel (*resp.*, base) categories in the target dataset.

After modifying the first step in each iteration, the final result (*i.e.*, $61.1\%$) only shows slight improvement against the result (*i.e.*, $60.9\%$) without mining base categories in target dataset. One possible explanation is that the mined pseudo bounding boxes are very noisy compared with the ground-truth bounding boxes in the source dataset, and thus simply adding the mined pseudo bounding boxes to the training set can only achieve slight performance improvement.

## 6 Hyper-parameter Analysis

Our method introduces three hyper-parameters, *i.e.*, $\alpha$, $\beta$, and $\gamma$ in Eqn. (7) and (8) in the main paper. Training SimNet also involves two hyper-parameters: $K$ and $M$ (see Section 3.3 in the main paper). To investigate the robustness of our method with these hyper-parameters, we vary each hyper-parameter in a certain range and plot the performance variance while keeping the other hyper-parameter fixed.

The results in Figure 3 and 4 demonstrate the insensitivity of our method to these hyper-parameters when setting them in reasonable ranges. For example, our method can generally achieve compelling results when setting $K$ (*resp.*, $M$) in the range of $[7, 10]$ (*resp.*, $[8, 10]$).

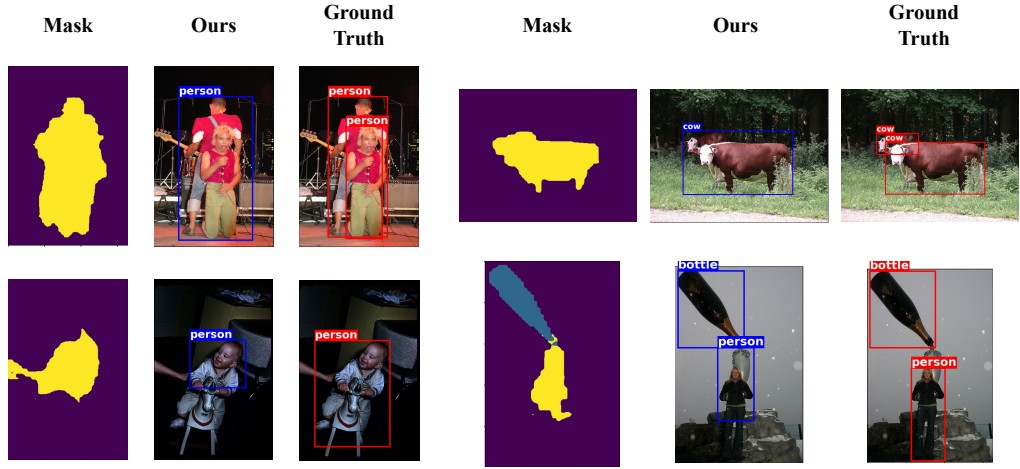

| Mask | Ours | Ground Truth | Mask | Ours | Ground Truth |

Figure 6: Illustration of some misleading masks .

# 7 Limitations of Our Work

## 7.1 Misleading Masks

In this paper, we introduce mask prior to help the object detection network identify and locate objects. However, the mask generator may generate incomplete or inaccurate masks, which would mislead the object detection network. Besides, the masks of the adjacent object instances from the same category cannot be separated, in which case the object detection network may detect the adjacent objects as an entire object. We display some example cases in Figure 6. In row 1, the joint region of the mask contains multiple instances, which induces the object detection network to detect only one object. In row 2, the incomplete mask misleads the object detection network to focus on the most discriminative part of the person.

## 7.2 Novel Categories with Poor Performance

Our method performs well on VOC-20 in general, while the performance on some novel categories is worse than some other approaches. Taking "*potted plant*" as an example, the best AP is $19.1\%$, which is lower than some WSOD methods, *e.g.*, CASD [1]. The poor performance might be caused by the large gap between novel category "*potted plant*" and base categories for that there is no base category similar to "*potted plant*". In this situation, transferring semantic similarity and mask prior becomes less effective.