# OpenReview forum: "Mixed Supervised Object Detection by Transferring Mask Prior and Semantic Similarity"
_NeurIPS.cc/2021/Conference — NeurIPS 2021 Poster_

### Official Review · Reviewer_ExAZ · 2021-07-14

**Rating:** 7
**Confidence:** 4

**Summary:**

This paper presents a Mixed Supervised Object Detection (MSOD) framework that is built on the Fast R-CNN framework and employs Multi-Instance Learning (MIL) to deal with weakly-supervised images. It adds two novel components to the underlying framework: Mask prior and Semantic similarity. Mask prior which is a semantic activation mask (e.g., CAM) works to enhance the ability to generate robust region proposals. Semantic similarity is used to remove inprecise proposals by checking its semantic similarity with other proposals. The proposed framework yields the better accuracy than other WSOD and MSOD methods on VOC 07 dataset.

**Ethical Concerns:**

I did not find any ethical concern from this work.

**Limitations And Societal Impact:**

I did not find any limitation or potential adverse societal impact.

**Main Review:**

This paper presents novel contributions in the methodology sufficient for consideration by NeurIPS community. The manuscript is very easy to follow.

My only concern is in experiments. Comparing with other baseline methods, most baselines used VGG16, but not the proposed method. This paper only provides the accuracy with VGG16 for the distrillation version. Therefore, I can not agree with that the proposed method has a benefit in the performance. I also wonder if other baselines also adopt multi-scale testing.

**Time Spent Reviewing:**

6 hours

---

> ### Author Response · Authors · 2021-08-10
> **Response to Reviewer ExAZ**
>
> **Q1**: My only concern is in experiments. Comparing with other baseline methods, most baselines used VGG16, but not the proposed method. This paper only provides the accuracy with VGG16 for the distrillation version. Therefore, I can not agree with that the proposed method has a benefit in the performance. I also wonder if other baselines also adopt multi-scale testing.
>
> **A1**:
> Thanks for your suggestion, we have conducted experiments on 'Ours' (VGG16 as the backbone without distillation)  and report mAP on VOC-20 test set and CorLoc on VOC-20 trainval set . Here are the results:
>
> mAP: 61.9%
>
> CorLoc: 76.8%
>
> These results are better than all baseline methods.
>
> By comparing 'Ours*' and 'Ours', we observe the minor difference between the performances of two backbones (mAP: 62.1% v.s. 61.9%, CorLoc: 77.1% v.s. 76.8%). This indicates that ResNet50 can not bring in significant improvement in mixed supervised setting, which is consistent with the observation in [28] in the main paper. As analyzed in [28], WSOD heads may back-propagate uncertain and erroneous gradient to backbones, whilst deep residual networks enlarge the erroneous information and deteriorate the visual representation learning.
>
> Besides, the default option is multi-scale testing in Table 1 and Table 2 in the main paper, so the baselines without marking '(single-scale)'  all use multi-scale testing.

---

> > ### Comment · Reviewer_ExAZ · 2021-08-29
> > **My concerns have been addressed well.**
> >
> > Thank you for the response. Most of My concerns has been addressed well. I will remain my initial rating. However, I agree with reviewer 3LCY to some extent, and therefore suggest that the authors develop an integrated end-to-end training strategy instead of a 3-step iterative training strategy. Based on my experience with other tasks it can significantly reduce training time without compromising accuracy.

---

### Official Review · Reviewer_itic · 2021-07-16

**Rating:** 6
**Confidence:** 4

**Summary:**

This paper introduces a mixed supervised object detection method with several improvements (mask generator/prior, semantic similarity transfer) proposed to improve the detection performance in the mixed supervision setting. The ability of detect novel categories is enhanced by the output of the mask generator that is supervised by only image-level labels. A MIL classifier is used to generate pseudo labels (detections) for the novel categories and the quality (average similarity scores) of the pseudo labels are determined by the semantic similarity network trained on only the base categories. Then, the pseudo labels are used in the detector's subsequent training and their losses are weighted by the average similarity scores obtained from the semantic similarity network. This encourages "good" pseudo labels to contribute more to the training and vice versa.

**Limitations And Societal Impact:**

They are almost adequately addressed. It would be good for the authors to relate the limitations of their method to some negative societal impact they may have.

**Main Review:**

Strengths
+ The ideas of using mask prior to enhance the detection performance on novel categories and semantic similarity transfer to weight the pseudo labels are quite interesting and novel in the context of mixed/weakly supervised object detection.
+ Strong experimental results in the standard setting. The proposed method outperforms state-of-the-art mixed- and weakly-supervised methods from very recent published papers.
+ Paper is generally well written, easy to follow, and has sufficient details for most parts of it.

Weaknesses
+ Lack of good explanation and evidence as to why the (metric-based) SimNet transfers well to novel categories. At the very least, the authors should show that the average similarity scores on base categories and "novel" categories are comparable, when that the scores are computed from the ground-truth boxes.
+ Table 3 does not specify clearly how the pseudo label weights are when no similarity function is used.
+ Sec 4.5 fails to mention which table it is referring to in the text.
+ Most of the weakly- and mixed-supervision prior works were focused on the small-scale dataset and splits. It would be very useful and more convincing if this paper could show its effectiveness in larger settings and datasets like Open Images and Objects365.

**Time Spent Reviewing:**

8

---

> ### Author Response · Authors · 2021-08-10
> **Response to Reviewer itic**
>
> **Q1**:Lack of good explanation and evidence as to why the (metric-based) SimNet transfers well to novel categories. At the very least, the authors should show that the average similarity scores on base categories and "novel" categories are comparable, when that the scores are computed from the ground-truth boxes.
>
> **A1**:
> To validate the transferability of similarity, we have evaluated SimNet on both base and novel categories, and reported precision, recall, and F1 scores in Section 2 of Supplementary. The results show that the gap between the performances of SimNet on base categories and novel categories is negligible.
> ***
>
> **Q2**: Table 3 does not specify clearly how the pseudo label weights are when no similarity function is used.
>
> **A2**:
> As mentioned in Line 225 in the main paper, the introduction of similarity enables the pseudo label weights w_i in Eqn.(1) in the main paper. Therefore, when no similarity function is used, the pseudo label weights w_i would be 1. We will make this clearer in the next version.
> ***
>
> **Q3**: Sec 4.5 fails to mention which table it is referring to in the text.
>
> **A3**:
> Thanks for pointing this out. Sec 4.5 refers to Table 3 in the main paper. We will fix it in the next version.
> ***
>
> **Q4**: Most of the weakly- and mixed-supervision prior works were focused on the small-scale dataset and splits. It would be very useful and more convincing if this paper could show its effectiveness in larger settings and datasets like Open Images and Objects365.
>
> **A4**:
> Thanks for your suggestion. Following previous works [21, 39] in the main paper, we use ILSVRC 2017, COCO 2017, and PASCAL VOC 2007 as our datasets. We will take the larger settings and datasets into consideration in the future research.

---

> > ### Comment · Reviewer_itic · 2021-08-21
> > **Response**
> >
> > Thanks. I am largely satisfied with the responses and will keep the positive rating. Please fix the paper issues if accepted.

---

### Official Review · Reviewer_gbMi · 2021-07-18

**Rating:** 7
**Confidence:** 4

**Summary:**

The paper proposes a method to do `mixed supervision` object detection, i.e. some part of the dataset is annotated fully while some part only has weak image level annotations.

The first contribution is to use semantic mask prior in object detection network. Prior works have already used semantic masks for weakly supervised detection, but the networks were separate. The paper proposes to integrate the mask prediction networks with the object detection network and use the masks to augment features for prediction candidate bounding boxes. The hope is that this would also generalize and produce better candidate boxes for target/novel categories.

The second contribution is to transfer semantic similarity. The paper incorporates a similarity (metric learning) network in the architectures and learns it using the bounding boxes of the base categories. Then the network is used to estimate the average similarities of pseudo bounding boxes for the novel categories. With the intuition that the average distance of a correct example wrt the rest of the class examples would be lower than that for an incorrect example, they use a weighted loss for object class prediction for the bounding boxes.

The training is done in an iterative manner and results are provided on public benchmarks COCO and ILSVRC datasets as source, and Pascal VOC 2007 as the target dataset.

**Main Review:**

It is specified that base and novel categories have no overlap, but is it also assumed that the weakly supervised images for novel categories would not have any of the base category objects? It seems to be so from the experimental settings too. This would be a strong assumption for deployment, and should be mentioned/discussed explicitly.

The contributions are novel but, the use of semantic maps has been done already before in weakly supervised settings with separate networks while the present work integrates the networks. This might be good or bad as the integrated network is trained only on the source dataset but a separate network might be trained on other data as well. Similar case for metric learning algorithm. However, showing that an integrated network works for the task and performs better than existing method is a good contribution.

For, the results table it seems the row Zhong et al.[39]+distill uses VGG16 as the backbone as there is no * on its side. Could you please clarify, what is the backbone? If VGG is used as the backbone for this setting, then for fair comparison the results with ResNet50 should also be reported as it is the best existing method.

A critical aspect in a mixed supervised detection setting is the amount of fully supervised data. An experiment showing the performance with varying amounts of supervised data would further provide an evaluation for the trade-off between supervised data and performance.

Overall the paper is well written and provides two contributions to the architecture as well as a tailored training method. The experiments reported are convincing.


**Time Spent Reviewing:**

3

---

> ### Author Response · Authors · 2021-08-10
> **Response to Reviewer gbMi**
>
> **Q1**: It is specified that base and novel categories have no overlap, but is it also assumed that the weakly supervised images for novel categories would not have any of the base category objects? It seems to be so from the experimental settings too. This would be a strong assumption for deployment, and should be mentioned/discussed explicitly.
>
> **A1**:
> In fact, we did not assume that weakly supervised images (i.e., target dataset) do not have the objects of base categories. To investigate the impact of base categories in the target dataset, we have conducted experiments to mine base categories in the target dataset in section 5 of the Supplementary. The results demonstrate that mining base categories in the target dataset only achieves slight improvement (i.e.,  61.1% mAP v.s.  60.9% mAP).
> ***
>
> **Q2**: The contributions are novel but, the use of semantic maps has been done already before in weakly supervised settings with separate networks while the present work integrates the networks. This might be good or bad as the integrated network is trained only on the source dataset but a separate network might be trained on other data as well. Similar case for metric learning algorithm. However, showing that an integrated network works for the task and performs better than existing method is a good contribution.
>
> **A2**:
> In this paper, we integrate mask generator, object detection network, and similarity network (these three networks share the same backbone). In our early experiments, we tried to use three separate networks, which only performs slightly better than the integrated network. For example, for 'Ours*(single scale)' on VOC-20 test set, using separate networks achieves 61.2% mAP and using the integrated network achieves 60.9% mAP. Considering training efficiency and model compactness, we adopt the integrated network as our framwork. Besides, following our iterative training strategy, the integrated network is trained on the source dataset only in the first iteration. In the following iterations,  the integrated network is trained on both source and target datasets.
> ***
>
> **Q3**: The results table it seems the row Zhong et al.[39]+distill uses VGG16 as the backbone as there is no '*' on its side. Could you please clarify, what is the backbone? If VGG is used as the backbone for this setting, then for fair comparison the results with ResNet50 should also be reported as it is the best existing method.
>
> **A3**:
> Yes, the backbone is VGG16.  Thanks for your suggestion. We report the results of ResNet50 (For simiplicity, we will not show the AP and CorLoc per category here. We will report the detailed results in our paper). The following results (i.e., mAP on VOC-20 test set and CorLoc on VOC-20 trainval set) of 'Zhong et al.[39]+distill*' are  directly copied from [39] in the main paper.
>
> mAP: 60.2%
>
> CorLoc: 75.2%
>
> These results are still worse than 'ours+distill*' (mAP: 62.9%, CorLoc: 77.7%) By comparing ’Zhong et al.[39]+distill\*' and ‘Zhong et al.[39]+distill’, we observe that mAP of 'Zhong et al.[39]+distill\*' is slightly better while CorLoc of ’Zhong et al.[39]+distill\*' even becomes slightly worse. This indicates that ResNet50 can not bring in significant improvement in mixed supervised setting, which is consistent with the observation in [28] in the main paper. As analyzed in [28], WSOD heads may back-propagate uncertain and erroneous gradient to backbones, whilst deep residual networks enlarge the erroneous information and deteriorate the visual representation learning.
>
> ***
>
> **Q4**: A critical aspect in a mixed supervised detection setting is the amount of fully supervised data. An experiment showing the performance with varying amounts of supervised data would further provide an evaluation for the trade-off between supervised data and performance.
>
> **A4**:
> It is interesting to study the amount of fully supervised data. Here we conduct some experiments and show the results with different sizes of the source dataset. Specifically, we randomly sample 20% and 50% of COCO-60 as the source dataset. The larger subset contains the smaller subset. We conduct experiments on 'Ours*(single scale)' and report mAP on VOC-20 test set . The final results after 4 iterations are shown below:
>
> 20%: 58.4%
>
> 50%: 59.8%
>
> 100%: 60.9%
>
> It can be seen that as the size of source dataset increases, the performance improves as well. We will add the results to our paper in the next version.

---

> > ### Comment · Reviewer_gbMi · 2021-08-21
> > **Keeping the favorable rating**
> >
> > Thank you for the responses. I would keep the favorable rating I had given initially. The results on different amounts of supervised data are interesting. It seems there is only an incremental effect after adding initial data. If you go below 20% then it would be interesting to see at what point the jump happens. Best,

---

### Official Review · Reviewer_3LCY · 2021-07-20

**Rating:** 6
**Confidence:** 4

**Summary:**

This paper studies mixed supervised object detection where some object categories are fully supervised and the rest are only labeled with class tags. It introduces a (1) weakly-supervised segmentation module and (2) a semantic similarity learning module to the existing framework to improve the performance. Experimental results on two benchmarks validated the effectiveness of the proposed framework.

**Ethical Concerns:**

I have read the Ethics Guidelines, and I don't find any concerns from this paper.
1. Potential negative societal impacts: It doesn't involve any social experiments, and doesn't use human-derived data.
2. General ethical conduct: the dataset used in this work are public ones.

**Limitations And Societal Impact:**

1. Limitations is discussed in the Appendix Sec.7. I think this section is pretty detailed and in-depth.
2. No societal impacts are discussed.

**Main Review:**

Strength:
- The problem setting is interesting. Compared to WSOD setting, the mixed-supervised task is more realistic and common in practice. However, this task is less explored than WSOD in the community.
- The experimental results are promising and both novel components help.
- Writing is mostly clear and easy to follow.

Weakness:
- To unifying the weakly-supervised segmentation and detection isn't a new idea. For example, multiple prior works [1-3] have studied and explored different architectures for this purpose. Some of these methods can be swapped into the proposed framework seamlessly and should be ablated and studied.


[1] Cyclic Guidance for Weakly Supervised Joint Detection and Segmentation. Shen et al. CVPR'19

[2] C-MIDN: Coupled Multiple Instance Detection Network With Segmentation Guidance for Weakly Supervised Object Detection. Gao et al., ICCV'19

[3] Multi-Evidence Filtering and Fusion for Multi-Label Classification, Object Detection and Semantic Segmentation Based on Weakly Supervised Learning. Ge et al., CVPR'18

- Sec.3.4 Iterative training seems to be complicated and hard to generalize. How important is this design choices? Will it become significantly worse if end-2-end training is adopted? I don't find experimental evidence to support this design choices.

- The learned mask is concatenated with the feature maps as input features. In fact, a more straight-forward solution to me is to multiply the feature map. I'm wondering whether this has been tried and how well does it perform.

- This paper follows the experimental setting from prior work where the datasets is split into non-overlap base and target categories. However, in practice the real problems are often more messy. The same class can have both annotation often the case. There are also prior work [4] studying this setting, which is missed in this paper. It would be interesting to see how well the proposed method performs to deal with mix-supervised for the same classes.

[4]. UFO2: A Unified Framework towards Omni-supervised Object Detection. Ren et al., ECCV'20


**Time Spent Reviewing:**

5

---

> ### Author Response · Authors · 2021-08-10
> **Response to Reviewer 3LCY**
>
> **Q1**: To unifying the weakly-supervised segmentation and detection isn't a new idea. For example, multiple prior works [1-3] have studied and explored different architectures for this purpose. Some of these methods can be swapped into the proposed framework seamlessly and should be ablated and studied.
>
> **A1**:
> In terms of using coarse semantic mask, our main contribution is cross-category transfer based on mask prior instead of unifying weakly-supervised segmentation and weakly-supervised object detection. As claimed in Line 57-59 in the main paper, we prove that the ability to detect objects based on mask prior and feature map can be transferred from base categories to novel categories, which will help detect the objects of novel categories.
>
> The mentioned papers [1*,2*,3*] do not have base/novel category split and thus do not involve cross-category transfer. They use coarse semantic mask to filter out noisy proposals or bounding boxes, without using ground-truth bounding box as supervision. In our framework, we require ground-truth bounding boxes of base categories as supervision to learn the ability to detect objects based on mask prior and feature map. Such ability can be transferred to novel categories. Therefore, the motivations and technical approaches of [1*,2*,3*] are considerably different from ours. Their methods cannot be directly embedded into our framework. Forcibly doing so will disrupt our framework and conflict with our motivation of cross-category transfer.
>
> We will cite and discuss  [1*,2*,3*] in the next version.
>
> [1*] Cyclic Guidance for Weakly Supervised Joint Detection and Segmentation. Shen et al., CVPR'2019
>
> [2*] C-MIDN: Coupled Multiple Instance Detection Network With Segmentation Guidance for Weakly Supervised Object Detection. Gao et al., ICCV'2019
>
> [3*] Multi-Evidence Filtering and Fusion for Multi-Label Classification, Object Detection and Semantic Segmentation Based on Weakly Supervised Learning. Ge et al., CVPR'2018
> ***
> **Q2**: Sec.3.4 Iterative training seems to be complicated and hard to generalize. How important is this design choices? Will it become significantly worse if end-2-end training is adopted? I don't find experimental evidence to support this design choices.
>
> **A2**:
> Following [39] in the main paper, we adopt an iterative approach to mine novel pseudo bounding boxes. Without such iterative training strategy, we cannot train Object Detection Network with the supervision of  pseudo novel bounding boxes. Previous works ([32,33,39] in the main paper) have proved the effectiveness of iterative approach. We have also shown the effectiveness of iterative approach in the Figure 5 of the Supplementary, in which the performance in the first iteration is significantly worse.
>
> Within each iteration, we have several steps as shown in the Algorithm 1 of the main paper. It is unsuitable to merge step 4, step 5 and step 6, because the input of each step depends on the output of previous step after convergence. We have tried merging step 5 and step 6, but the network is hard to converge and the performance is significantly degraded.
>
> ***
> **Q3**: The learned mask is concatenated with the feature maps as input features. In fact, a more straight-forward solution to me is to multiply the feature map. I'm wondering whether this has been tried and how well does it perform.
>
> **A3**：
> Thanks for your suggestion. For element-wise multiplication, we calculate the max value of mask over the category dimension, and then multiply the mask and the feature map. We conduct experiments with 'Ours*(single-scale)' and report mAP on VOC-20 test set. The result of element-wise multiplication is 60.4%, which is worse than the results of concatenation 60.9%.
> ***
> **Q4**: This paper follows the experimental setting from prior work where the datasets is split into non-overlap base and target categories. However, in practice the real problems are often more messy. The same class can have both annotation often the case. There are also prior work [4] studying this setting, which is missed in this paper. It would be interesting to see how well the proposed method performs to deal with mix-supervised for the same classes.
>
> **A4**:
> The mentioned paper [4*] investigated the case that all categories have different forms of annotated data, which is quite different from ours. In our work, we aim to use the off-the-shelf fully-annotated object detection datasets to help detect the categories which have only weakly-annotated data, and thus we split the categories into base categories and novel categories. Our main goal is transferring from base categories to novel categories.
>
> In some real-world cases, a few categories may have both fully-annotated data and weakly-annotated data. Therefore, we conduct the experiments with our proposed method under this setting. In our original COCO-60 dataset, we removed all the images which contains the objects of novel categories following [39] in the main paper. Now among the 20 novel categories, we choose the first 10 novel categories based on the alphabetical order as mixed categories, and treat the remaining 10 categories as novel categories. Then, we only remove the images which contains the objects of 10 novel categories from COCO and name it COCO-70 instead of COCO-60 dataset, so that 10 mixed categories have both fully-annotated data in the source dataset and weakly-annotated data in the target dataset. We conduct experiments with 'Ours+distill*', and report  mAP on VOC-20 test set and CorLoc on VOC-20 trainval set.
>
> The results are shown below. It can be seen that mAP and CorLoc of mixed categories are much higher than novel categories thanks to the extra supervision of bounding boxes of mixed categories.
>
>  ———————————————————
>
> ｜　　　　　　　　 ｜mAP ｜  CorLoc  ｜
>
>  ———————————————————
>
> ｜mixed categories:｜  69.9 ｜　86.3　｜
>
>  ———————————————————
>
> ｜novel categories:	｜  62.4   ｜　75.1　｜
>
>  ———————————————————
>
>
> [4*] UFO2: A Unified Framework towards Omni-supervised Object Detection. Ren et al., ECCV'20.

---

> > ### Comment · Reviewer_3LCY · 2021-08-30
> > **Response**
> >
> > Thanks for the detailed reply. It answers most of my questions. Please consider incorporating some of the stuff here into the future version, and further simplify the framework. Raised my score to 6.

---

### Decision · Program_Chairs · 2021-09-27

**Decision:**

Accept (Poster)

**Comment:**

The reviewers have discussed the paper and are generally positive. In particular, some of the concerns raised in the reviewer were lifted, which resulting in improving the scores. I would like to encourage the authors to incorporate the suggestions into the final version and recommend accepting the paper.